# Ambient Intelligence to Improve Construction Site Safety: Case of High-Rise Building in Thailand

**DOI:** 10.3390/ijerph17218124

**Published:** 2020-11-03

**Authors:** Kriengsak Panuwatwanich, Natapit Roongsrisoothiwong, Kawin Petcharayuthapant, Sirikwan Dummanonda, Sherif Mohamed

**Affiliations:** 1School of Civil Engineering and Technology, Sirindhorn International Institute of Technology, Thammasat University, Pathum Thani 12120, Thailand; kriengsak@siit.tu.ac.th (K.P.); 5822800116@g.siit.tu.ac.th (N.R.); 5822781217@g.siit.tu.ac.th (K.P.); 5822780789@g.siit.tu.ac.th (S.D.); 2School of Engineering and Built Environment, Griffith University, Gold Coast, QLD 4222, Australia

**Keywords:** construction sites, high-rise, falls, ambient intelligence, human behavior

## Abstract

The relatively high rate of injuries in construction is not surprising, as site work by its very nature ranks highly on fundamental risk factors. Working at heights often magnifies these risk factors. The literature reveals that falls from heights accounts for a large percentage of injuries in construction worldwide. Thailand is no exception, where fall accidents constitute the majority of high-rise construction accidents despite preventive measures being implemented. This paper examines how the use of a simple Ambient Intelligence (AmI) system—a device comprising a microcontroller, microwave sensors, Light Emitting Diode (LED) and audio alarm—could help to affect safety behavioural change of on-site construction workers in order to decrease the potential for fall accidents. An experiment was conducted at a high-rise building construction site in Bangkok, Thailand to examine the effectiveness of the AmI in helping workers mitigate the risk of falling from heights. The analysis of the data collected over two work weeks from the pre- and post-AmI application using X-bar charts and one-way analysis of variance (ANOVA) revealed a significant reduction of about 78% in the number of workers passing through the fall hazard zones. The finding established the potential of a simple AmI for reducing the risk of fall accidents.

## 1. Introduction

### 1.1. Falls from Heights

Globally, the construction industry continues to strive to improve workers safety as they typically perform physically demanding tasks in poor work conditions (e.g., extreme temperature, excessive noise, confined space) and are frequently exposed to work-related hazards (e.g., falling from a height, struck by equipment, falling materials or objects) [1]. Tragically, occupational accidents in this industry are frequent, and may lead to permanent disability and a high rate of fatalities [2]. The cost of fatalities arising from unsafe working environments and actions goes far beyond the loss of workers’ lives, imposing broader economic and psychological costs [3].

Falls from a height (FFH) still consistently have the highest rates amongst construction accidents compared with others types of accidents (e.g., vehicle collisions, hits by moving or falling objects, being trapped between stationary and moving objects and contact with electricity), and when compared to other industries’ accidents [4]. In 2016, 697 fatal FFHs occurred at workplaces across all industries in the US, with 53% of these fatal accidents taking place in construction [5]. More recently, in the United Kingdom, it was reported that FFH accounted for 25% of all fatal injuries, with half of all FFH deaths over the last five years in the construction sector [6]. In Thailand, and according to the National Statistics Office (NSO), FFHs are the main cause of fatality among construction workers [7]. The most common factors associated with FFH are risky activities, individual characteristics, site conditions, organizational characteristics, agents (i.e., scaffolds/ladders) and weather conditions [8]. Consequently, any attempt to minimise the risk of FFH would result in saving lives.

### 1.2. Human Behaviour and Situational Awareness

As stated previously, the most common factors associated with FFH are risky activities and individual characteristics. Therefore, it is not surprising that understanding human behavior has attracted several researchers’ attention. Researchers have argued that human behaviour accounts for most accidents and so understanding and being able to modify behaviour should be crucial to improving safety performance. Although behaviour, and by extension, safety behaviour, is influenced by activators/antecedents and consequences [9], unsafe behaviours are in the individual’s control and also within the scope of supervisors and management to control effectively [10]. It could be argued that if one does not recognise that there is a risk, then one may not act appropriately. Reasons to account for why humans behave in dangerous and careless ways is an attractive research area. However, it is acknowledged that fluctuation of risk perception amongst individuals makes it difficult to identify the causes, effects and prevention techniques for risk-taking behaviour [11].

In the context of FFH, how well are the risks associated with falling from heights perceived? Situational awareness and risk perception of workers are key factors that have a direct influence on falls [12]. Inadequate risk perception contributes to accidents, in that workers recognise the hazard but do not modify their behaviour accordingly [13]. Misjudgements could arise from a lack of situational awareness or knowledge provided by supervisors [14]. Situational awareness could be defined as being aware of what is happening around you in terms of where you are, where you are supposed to be, and whether anyone or anything around you is a threat to your health and safety. In short, it is knowing what you are doing, what you should be doing, what should be happening as you are doing it and what should eventually happen when it is done.

Situational awareness is a three-step process which includes detection of hazardous signals, perception and comprehension of the risks associated with the hazard, and projection of the consequences associated with the decision options [15]. Consequently, an important component of situational awareness is a worker’s ability to accurately perceive risks. Risk perception is fundamental because, even if hazards are identified, workers may involuntarily behave unsafely when they inaccurately perceive and value risk.

Attempting to change behaviours can prove challenging especially when the safe behaviour to be engaged in is not known. Geller [16] places behaviour into three classes; (1) behaviours that are directed by others, (2) behaviours that are directed by one’s self, and (3) behaviours that are not directed consciously but based on reflex. The latter class is also known as innate behaviour, which can be defined as an involuntary and rapid response to a stimulus. In this paper, the stimulus is being provided by a sensitive adaptive electronic environment as explained below.

### 1.3. Ambient Intelligence

The term “ambient intelligence” (AmI) is about sensitive adaptive electronic environments that respond to actions of persons, objects and cater for their needs [17]. Given that communication, interaction and information exchanges among people and computer systems have become more efficient as technology offers improvements to life quality and services, the presence of AmI technology provides the ability of passive data collection, thus reducing the physical effort required to complete a task that involves repetition and/or supervised actions [18]. Over the past few years, AmI applications have been reported in many fields of study. For instance in healthcare where AmI is used for the tele-monitoring of Alzheimer’s patients subject to risk of heart failure [19] or as a home-based tele-monitoring system to provide constant observations of patient’s blood pressure, oxygen saturation and weight [20]. In tourism, AmI brings intelligence to tourism ecosystems and makes those systems sensitive, flexible and adaptive to stakeholders’ needs [21]. As the AmI environment is characterised by the merging of physical and digital space, AmI has also huge potential for enhancing educational activities [22], manufacturing processes [23] and logistics services [24].

To prevent construction-related accidents, safety supervisors need to monitor the construction site conditions thoroughly. Based on the site observation, actions can be made to protect workers under safety risks [25]. However, it is impossible to continually monitor site operations to detect unsafe behaviours and/or conditions without the help of technologies [26]. This fact has led to several attempts to apply sensing technologies in order to identify workers under safety risks with various sensor-based technologies being adopted for construction safety management [27,28]. For example, Riaz et. al. [29] developed a prototype utilising BIM to present data received from wireless sensors placed in a construction site and reported its potential to facilitate intelligent monitoring of worker safety in confined spaces through real-time sensor data to avoid time sensitive emergency situations typically encountered by workers operating in such work environments. Liu et al. [30] integrated Building Information Modelling (BIM) and an indoor positioning system inertial measurement unit (IPS-IMU) to create an inspection and auto-warning system for onsite workers, foremen and safety managers that enable real-time alerts when workers are exposed to fall hazards. Through a case study, Jiang et. al. [31] developed and implemented a cyber-physical system (CPS) for safety management in smart construction site, which showed a reduction of on-site workers being close to the construction hazard areas.

The presence of the above technologies makes possible the development of an AmI system that can be used to improve construction site safety. AmI, as proposed herein, refers to a digital environment that proactively supports workers working at height, where its functionality is based on the ability to use many inter-related technologies. Despite many existing AmI-enabling technologies however, there has been very limited empirical research attempting to examine how an AmI system can be developed and applied to affect positive change in workers’ safety behaviour at a construction site. Therefore, the research presented in this paper was aimed to examine the efficacy of a simple AmI system in reducing the risk of FFH. The purpose of the AmI was to improve the worker’s awareness of areas where fall hazards exist. This is achieved by using an alarm triggered by microwave sensors to warn workers when they are in the proximity of a fall hazard area, which may not be clearly visible or obvious in various site conditions. The outcome from this research would be of interest to developing countries seeking to implement a simple AmI system constructed from generally available technologies to improve construction workers’ safety behaviour.

## 2. Materials and Methods

Due to the nature of this present study, an on-site experiment was designed and implemented to explore AmI’s potential in decreasing FFH. The experiment involved an installation of a simple AmI system at a high-rising building construction site in Bangkok, Thailand to collect and anlayse the data before and after the implementation of the AmI system. The following sections describe the AmI system used, the experiment site and the data collection process.

### 2.1. The AmI System

A simple AmI system consisting of a set of microwave sensors (Figure 1) and a microcontroller (Figure 2) was specifically constructed for this research. It is worth noting that, initially, a typical security camera with an image-based motion detection capability was considered. However, it was found that image-based motion detection can be quite sensitive to any object crossing the view of the camera. Given that the cameras would have to be installed in the construction site environment, the interference from dust, particles and various objects was a concern. In addition, it was determined that the developed AmI system must be based on simple and robust technologies. Compared with one equipped with sophisticated technologies, such a simple and robust system would be more appropriate for the context of a developing country, which could, in turn, increase the adoption of technologies to improve construction safety in the industry.

The developed AmI system can detect a movement within a specific area and sound the alarm whenever the movement is detected. It can also collect and store the frequency of movement detection in a memory card, which can then be transferred to a computer for further analysis. As the name implies, the sensors utilise microwaves as a medium for motion detection. Each sensor emits a microwave which on reflection off the contact surface, changes its wavelength according to the type of contact surface, and gets detected by the receiver. A microwave sensor has adjustable wave intensity for suitable sensitivity, with the scale ranging from 1–7. An intensity of 1 is capable of detecting a large solid object in motion with a range of approximately 3–4 m and an area of 0.5 × 0.4 m. The maximum intensity of 7 allows a detection range of 6–7 m with an area of 3 × 1.5 m. It is worth nothing that the sensor cannot distinguish between a single object and multiple objects. The microwave emitted by a sensor can penetrate several types of thin materials (e.g., paper, plastic, small insects and water); thus, the sensor will not detect such items. The microwave, however, does not penetrate such materials as steel, aluminium, stone and other rock-like materials.

The components of the developed AmI system are presented in Table 1, whereas the layout of these components is illustrated in Figure 3.

### 2.2. Experiment Site and Setup

To examine the potential of AmI in improving construction site safety and particularly to help lower FFH risk, a 23-storey building construction site in central Bangkok, Thailand was chosen for this study to conduct the experiment. At the time of the experiment, the superstructure was already completed and the Mechanical, Electrical and Plumbing (MEP) work was underway. Upon consultation with the project manager and the site engineer, the landing platforms between the building and the passenger hoistway were identified as a hazardous location, suitable for the experiment where FFH risk could be identified from the gap and opening on the sides of the platforms (Figure 4) especially during peak working hours with high traffic of workers entering and exiting the hoist lifts. Although guardrails were already installed at both sides of each landing platform to prevent fall accidents and the gap may appear to be small, the study was deemed beneficial from the actual site condition where workers’ safety behaviour could be best observed. Therefore, the sensors were placed at the portal frames adjacent to the landing platform (Figure 5) and positioned to detect any object within both sides of the lift landing platforms on the 14th story as shown in Figure 6. It should be noted that only the hoist lift serving Platform 2 was in operation for the observed floor during the period of experiment (as explained in the next section). The intensity of the sensors was set to 2–3 as the distance from the position of the sensors to the detection areas was not far. When a motion is detected, the sensor counts as 1 with associated time-stamp recorded by the data logger and stored in the SD card. A hard reset button on the microcontroller was pressed in if the data logger failed to log the detection due to possible interference or device malfunction. Prior to the experiment, the system was tested at the site over a period of four working days to ensure that it functioned properly and to observe any factors that may affect the operation of the sensors.

### 2.3. Data Collection Process and Analysis

To examine the efficacy of the AmI system in improving construction site safety by assisting workers to avoid FFH risk areas, the data collection process was broken down into the pre- and post-AmI deployment. In the pre-AmI deployment stage, only the warning alarm of the system was turned off, whereas the sensors were activated to count the number of workers passing through the designated hazardous areas. That is, the workers were not warned by the system when moving into the areas and the observation, in this stage, was effectively made under normal working conditions (i.e., without the AmI system). In the post-AmI deployment stage, the warning alarm was turned on. When a worker moved into the designated hazardous areas detectable by the sensors, the audio and LED alarm would go off. On the first day of the AmI deployment stage, however, the research team explained clearly to all site workers the purpose of the alarm and highlighted the designated areas with a high risk of FFH.

The duration of the experiment was determined in conjunction with the site engineer to ensure that it represented the normal working load for the MEP work on the observed floor. In other words, the duration was chosen such that it did not fall within a period of an unusually high workload (e.g., a project schedule crashing/compressing) or a low workload (e.g., long public holidays). As a result, the experiment was conducted over a period of two working weeks with the pre-AmI stage taking place in the first week (Monday 22 April–Friday 26 April 2019) and the post-AmI stage in the second week (Monday 29 April–Friday 3 May 2019), with the data collected at one hour intervals from 8:00 a.m. to 7:00 p.m. daily. The daily count data were also normalised for subsequent analysis based on the total number of workers working at the observed location as provided by the project manager. This is to account for the possible effect of the varying total number of workers at the site on the daily count data recorded by the sensors.

Data analysis was primarily conducted in order to determine whether there was any significant difference between the number of workers passing through the designated hazardous areas before and after the deployment of the AmI system. The Statistical Process Control (SPC) chart was firstly employed to examine the trends of the collected data. The X-bar chart was considered appropriate as it deals with variable (as opposed to categorical) data that involve a data sample (i.e., day of the week) with multiple observations in the sample (i.e., daily hourly detection count). Typically, the X-bar chart (and other SPC charts) will include the upper and lower control limits that can be used to indicate whether the variation of the data is statistically in control. For this research, the X-bar chart was used to detect any significant upward/downward trends, which indicate a significant increase or decrease in the number of risks attempted by the workers (as represented by the normalised daily count). This significant change between the pre- and post-AmI deployment data was further ascertained using One-Way Analysis of Variance (ANOVA). The results of both analyses are presented in the following sections.

## 3. Results

### 3.1. Pre- and Post-AmI Deployment Data

Following the experiment setup and the one-week test-and-run period mentioned in Section 2, the pre-AmI deployment stage began on Monday 22 April and continued through to Friday 26 April 2019. It is worth reiterating that in this phase, the visual (LED) and sound alarms were turned off to observe the workers’ behaviour in their normal working state—no matter how many workers stood in the designated hazardous area, they would not notice. The post-deployment stage, with the visual and audio alarms turned on, followed on from Monday 29 April–Friday 3 May 2019.

Initial data screening indicated that logging errors occurred on Monday 22 April and Friday 3 May 2019. As a result, the data obtained for these two dates were removed from the analysis. Table 2 presents the hourly count data obtained from the pre- and post-deployment stages. It also shows how the daily total number of workers reported to work at the site varied throughout the same period, giving rise to the need for data treatment (i.e., normalisation of daily count data). This was simply achieved by dividing the daily count data by the associated total number of workers at the site on that day. For each stage, the normalised daily count data are provided in Table 3, along with the corresponding daily average and range values (needed for producing the X-bar chart).

### 3.2. X-bar Chart Results

To examine the efficacy of the implemented AmI device, the X-bar chart was used to detect any significant change in the trends of the number of FFH risk attempts by workers (as represented by the normalised daily count) pre- and post-AmI deployment. To construct an X-bar chart, the following parameters were determined as follows:

X-bar ($\bar{x}$) values: The average values of the normalised daily count (from Table 2)X-double-bar ($\overset{=}{x}$) values: The average of the X-bar (=0.674)R values: The range of normalised daily count (from Table 2)R-bar ($\bar{R}$) values: The average of the R values (=0.819)A_2_ value: This is a standard estimated value based on the number of observations. For X-bar chart with 8 hourly counts per day, the value A_2_ is equal to 0.37 at 3σ control limits.Upper Control Limit (UCL) = $\overset{=}{x}$ + A_2_
$\bar{R}$ = 0.977Lower Control Limit (LCL) = $\overset{=}{x}$ − A_2_
$\bar{R}$ = 0.371

Based on the above parameters, the X-bar chart was plotted as presented in Figure 7. According to the figure, there is a significant drop in the normalised daily count (i.e., risk attempts per person) during the post-AmI deployment. Prior to its deployment, however, there were three days with reported risk attempts above the UCL (approximately 1.27 risks attempted per person) representing baseline risk behaviour during any typical workday. After the AmI was deployed, the trend of the risk attempts dropped significantly (below the LCL) to an average of only 0.25 risk attempts per person. This finding supports the notion that the AmI was effective in reducing risky behaviour. It should be noted that both control limits, in this research, may not be strictly interpreted in the same manner as what is commonly used in manufacturing, where process variation is largely driven by machinery. In the manufacturing industry, most production aspects can be precisely controlled and observed. This is, however, not the case for construction health and safety monitoring where human behaviour coupled with continual process changes resulted in highly variable data. Both control limits used in this research provide the expected variation of counts over a workday. When the recorded data are beyond either control limit, it simply means that they have exceeded the expected range. To further verify that the reduction of the risk attempts is not due to chance (i.e., statistically significant), further statistical analysis using the One-Way Analysis of Variance (ANOVA) was required as explained in the next section.

### 3.3. ANOVA Results

The ANOVA compares means between groups of interest and determines whether any of those means are statistically significantly different from each other [32]. In this research, the comparison was made between all the hourly count data (not the daily average as used in the X-bar chart) in the pre- and post-AmI deployment stages. The analysis was conducted using the Minitab software. The results are presented in Table 4.

According to Table 4, the key parameters from the ANOVA analysis are the *p* value and the Tukey pairwise comparison. The *p* value, or calculated probability, is the probability of finding the observed, or more extreme, results when the null hypothesis (i.e., no difference between the normalised hourly count data between pre- and post-AmI) is true. A *p* value of two groups less than a pre-chosen probability or significance level (α = 0.05 in this research) is considered statistically significantly different, and this implies that there is a 0.05 (or 5%) probability that the null hypothesis is true. In this case, it is judged that the null hypothesis “all means are equal” is not acceptable and the two groups are statistically significantly different. The Tukey statistic with a 95% confidence interval compares means between pre- and post-AmI deployment. If the interval of the means does not contain 0, then they are judged to be significantly different [32]. Moreover, grouping information using the Tukey method illustrates different groups between the pre- and post-AmI stages. To conclude, the ANOVA reinforced the efficacy of the AmI as the analysis verified the reduction in the mean values of the normalised hourly counts from 1.1020 in the pre-AmI deployment stage to 0.2460 (approximately 78% reduction) in the post-AmI deployment stage (as shown in the Tukey pairwise comparisons) is statistically significant (at 95% confidence level).

## 4. Discussion

Safety in construction sites remains a high priority for every company, yet the number of accidents and casualties persist. Statistically, falling from heights is the most common cause of death in construction sites, but can be eliminated or reduced by an implementation of preventative and protective site safety measures. Protective site safety measures are commonly seen, for example, through the installations of fall protection and the use of personal protective equipment (PPE), while prevention could focus on instilling a good safety culture through, for example, the use of sign boards and routine safety meetings. This research focuses on reducing the risk of fall accidents at construction sites using a preventative warning system aimed at assisting workers to keep away from hazardous areas.

Subject to some limitations, the results and analysis of the pre- and post-AmI deployment stages are promising as the number of daily counts was notably reduced. However, visual inspection of the data alone cannot prove whether AmI affects workers’ behaviour, or whether this was a coincidence. Hence, an X-bar chart and one-way ANOVA were used to verify the statistical significance of the improvement as a result of the AmI implementation. As shown in the results, the data exceeded the upper and lower control limits, reflecting the fact that the data exceeded the expected variation range. In this instance, post-AmI data moving below the lower control limit is a very positive result. The drop in the number of risk attempts per worker was also reinforced statistically via one-way ANOVA. The estimated *p* value was less than the pre-chosen probability or significance level (0.05), implying the rejection of the null hypothesis that “all means are equal” and accept the alternative hypothesis (“not all means are equal”). In addition to the *p* value, the Tukey difference of means analysis indicated that the difference between means was not 0 and the data groups were different; thus, the results were statistically significantly different. The implementation of AmI was proven to provide a real impact on construction site safety. In addition to physical preventative safety measures, AmI offers an augmenting method to improve construction site safety for companies to consider. Although it may not have the ability to completely replace the traditional site safety protections, it could further improve construction site safety by altering human behaviour and awareness of risk. Since the attitude of workers towards safety behaviours contributes to hazardous actions [33], every person must make constant efforts to learn and understand the risks in their work areas in order to develop risk awareness habits [34]. The AmI could prove to be an effective mechanism in shaping such awareness.

To enhance the results further, the adage of “old habits die hard” must be observed. Since the number of observations (hourly observations) and the sample (observed dates) were limited, there is a risk that workers ignore the alarm and over time resume their risky behaviour. Research conducted by Stewart et al. [35] indicated that people inherently resist change over long term periods and revert back to their accustomed behaviour. A short period of observation may show signs of change; however, this may not continue into the future without adequate reinforcement measures. In theory, people have to be taught and educated on the correct safe behaviour to engage in and, with repetition, the new behaviour should move from being directed by others through to self-directed and eventually automatic. Once the behaviour is automatic, it is likely to be sustained. Geller [16] argues that it is sometimes good for behaviours to occur out of habit, however it is better to engage in most safety related behaviours after some form of cognitive reasoning because situations vary and therefore solutions may have to be adapted.

## 5. Conclusions

In this paper, a simple but effective AmI device has been designed and developed as a tool to decrease the potential for fall accidents at a construction site. The device was applied to a real high-rise building construction site in Bangkok, Thailand, and was tested to examine its effectiveness in helping workers mitigate the risk of falling from heights. The results indicate that the device has the potential to improve workers’ behaviours when working in areas with a high risk of FFH. The X-bar chart shows a clear reduction in the average normalised daily counts of the number of workers passing through the fall hazard areas (i.e., risk attempts). The results from the ANOVA also confirmed the statistical significance of this reduction.

Understanding the degree of severity when working in high-elevation zones will greatly reduce the potential for falls; hence, developing an awareness around FFH risk-prone areas should be institutionalised. Providing pre-education and training only mitigates a moderate amount of perceived risk, since such knowledge varies by location and environmental conditions, among others. The application of AmI could therefore provide an effective augmented safety prevention mechanism by reinforcing and sustaining such risk awareness required throughout the entire construction project.

It is important to note that the findings from this study be interpreted with some reservation due to several limitations. Experimentally, the period of the observation was relatively short as there was an administrative restriction making the longitudinal study impossible. As pointed out in the previous section, there is a possibility that in the longer term, the workers may gradually ignore the alarm and revert to their previous behaviour. Therefore, an extended period of observation is required in future studies to fully assess the longer-term effects of AmI. Technically, due to the study being conducted in the context of a developing country, the aspect of “intelligence” of the AmI system employed in this study is limited to only an audio alarm, which could be considered too simplistic with respect to existing technologies. Future studies could benefit from integrating different types of technologies that enable the AmI the collection of various types of data, such as images, location and temperature, which could allow for a more comprehensive data analysis, on-site monitoring and the provision of a better feedback mechanism to the workers and managers.

## Figures and Tables

**Figure 1 ijerph-17-08124-f001:**
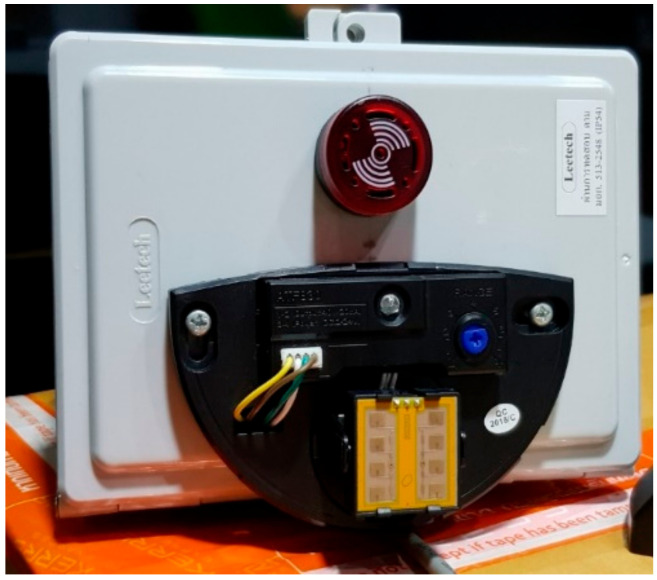
Microwave sensors.

**Figure 2 ijerph-17-08124-f002:**
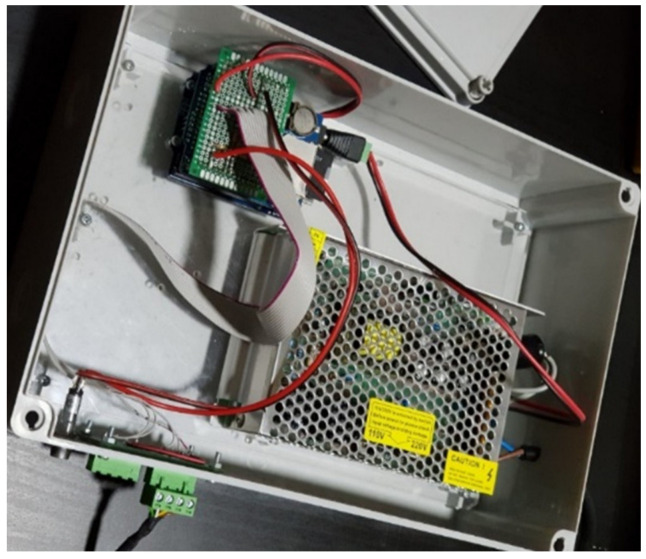
Microcontroller.

**Figure 3 ijerph-17-08124-f003:**
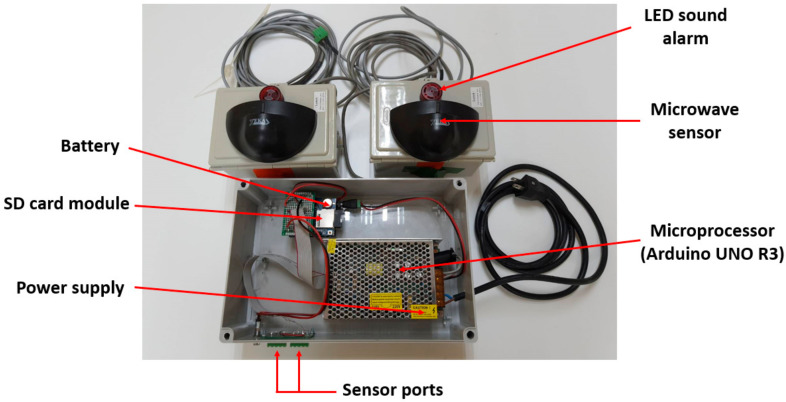
Layout of the AmI system components.

**Figure 4 ijerph-17-08124-f004:**
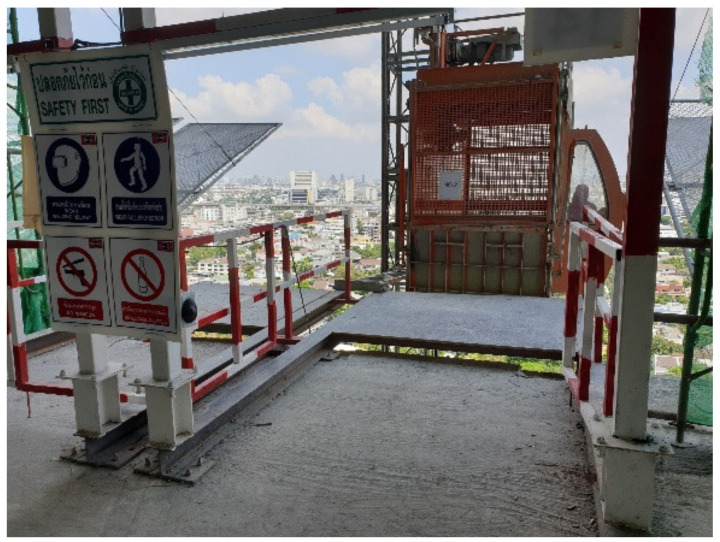
Landing platforms for hoist lift.

**Figure 5 ijerph-17-08124-f005:**
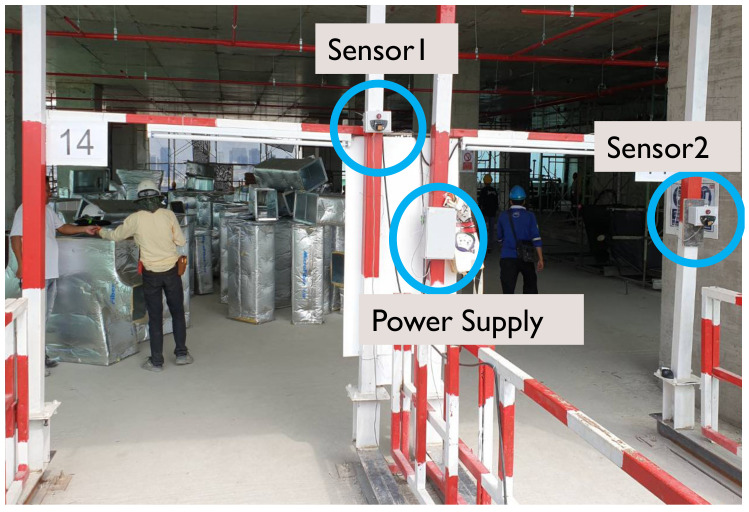
Sensors location.

**Figure 6 ijerph-17-08124-f006:**
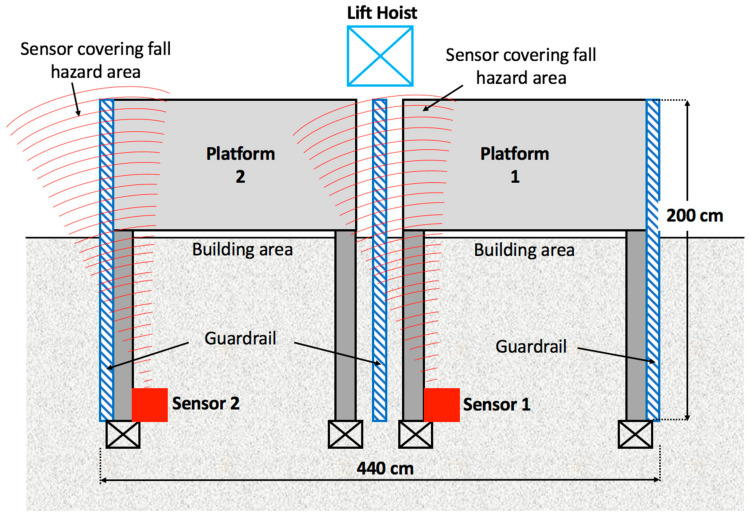
Sensor detection areas (top view).

**Figure 7 ijerph-17-08124-f007:**
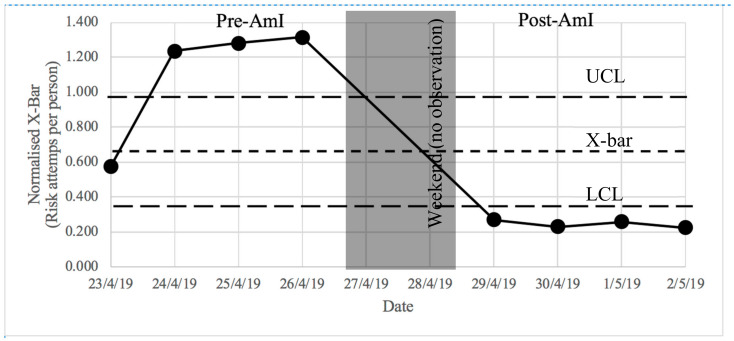
X-bar chart.

**Table 1 ijerph-17-08124-t001:** System components.

Component.	Quantity	Function
Microcontroller (Arduino Uno R3)	1	Serve as a processing unit, write data and variables in the storage, data logger
16 GB memory card	1	Main storage
Memory card module	1	Reading and writing data into the memory card
Real time clock module	1	Real-time clock
Power Supply 12 V 5 A	1	Convert 220 V 5–15 A to 12 V 5 A and supply power to device
Light Emitting Diode (LED) and Sound Alarm	2	Provide visual and audio alarm when a motion of an object is detected
Microwave Sensor (WB8300)	2	Detect object in motion
Custom Input Port	2	Custom-made input port for data transfer on data logger

**Table 2 ijerph-17-08124-t002:** Daily count data from the pre- and post-AmI deployment stages.

Starting Time	Pre-AmI Deployment Daily Count (23–26 April 2019)				Post-AmI Deployment Daily Count (29 April 2019–2 May 2019)			
	23/4/19	24/4/19	25/4/19	26/4/19	29/4/19	30/4/19	1/5/19	2/5/19
8:00	34	318	361	199	70	40	32	58
9:00	92	300	252	333	57	67	14	58
10:00	30	210	272	306	53	48	16	48
11:00	150	284	357	295	87	125	48	108
13:00	65	199	248	140	45	75	63	66
14:00	89	147	233	130	36	30	7	12
15:00	110	130	254	287	26	24	12	9
16:00	302	313	320	498	66	54	14	1
Total number of workers at site	190	192	224	208	204	251	100	199

Note: AmI = Ambient Intelligence.

**Table 3 ijerph-17-08124-t003:** Normalised data from the pre- and post-AmI deployment stage.

Starting Time	Pre-AmI Deployment Normalised Daily Count (23–26 April 2019)				Post-AmI Deployment Normalised Daily Count (29 April 2019–2 May 2019)			
	23/4/19	24/4/19	25/4/19	26/4/19	29/4/19	30/4/19	1/5/19	2/5/19
8:00	0.179	1.656	1.612	0.957	0.343	0.159	0.320	0.291
9:00	0.484	1.563	1.125	1.601	0.279	0.267	0.140	0.291
10:00	0.158	1.094	1.214	1.471	0.260	0.191	0.160	0.241
11:00	0.789	1.479	1.594	1.418	0.426	0.498	0.480	0.543
13:00	0.342	1.036	1.107	0.673	0.221	0.299	0.630	0.332
14:00	0.468	0.766	1.040	0.625	0.176	0.120	0.070	0.060
15:00	0.579	0.677	1.134	1.380	0.127	0.096	0.120	0.045
16:00	1.589	1.630	1.429	2.394	0.324	0.215	0.140	0.005
Average normalized daily count	0.574	1.238	1.282	1.315	0.270	0.231	0.258	0.226
Range of normalised daily count (max − min)	1.432	0.979	0.571	1.769	0.299	0.402	0.560	0.538

Note: AmI = Ambient Intelligence.

**Table 4 ijerph-17-08124-t004:** ANOVA results.

One-way ANOVA.					
**Method**					
Null hypothesis			All means are equal		
Alternative hypothesis			Not all means are equal		
Significance level			α = 0.05		
Equal variances were assumed for the analysis.					
**Analysis of Variance**					
Source	DF	Adj SS	Adj MS	F-Value	*p* Value
Stage	1	11.725	11.7252	81.27	0.000
Error	62	8.945	0.1443		
Total	63	20.670			
**Model Summary**					
S.	R-squared			R-squared(adjusted)	
0.379832	56.73%			56.03%	
Means					
Stage	N	Mean	StDev	95% Confidence Interval	
Post-AmI	32	0.2460	0.1505	(0.1117, 0.3802)	
Pre-AmI	32	1.1020	0.5156	(0.9678, 1.2362)	
Pooled StDev = 0.379832					
**Tukey Pairwise Comparisons**					
Grouping Information Using the Tukey Method and 95% Confidence					
Stage	N		Mean	Grouping	
Pre-AmI	32		1.1020	A	
Post-AmI	32		0.2460	B	

Note: ANOVA = Analysis of Variance, AmI = Ambient Intelligence, Adj SS = Adjusted sum of squares, Adj MS = Adjusted mean squares, DF = The total degrees of freedom, StDev = Standard Deviation.

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
