# Peer review of "Ambient Intelligence to Improve Construction Site Safety: Case of High-Rise Building in Thailand"

_ijerph, 2020, doi:10.3390/ijerph17218124_

Round 1
Reviewer 1 Report
I very appreciate the authors’ effort to help constructors to reduce the risk of falling from heights (FFH). And I believe this system can warn people who access some danger areas, and reduce the potential risks. If it was an undergraduate student project demo in an exhibition, I definitely will recommend it. However, as a research paper, it looks not prepared well to be published.
- First of all, this paper is lack of originality and novelty. The microwave sensors (or Radar in generally) have been widely applied in motion detection. It was one of the dominant technologies to detect people access a certain area decades years ago before security cameras and computer vision technology developed. And still be widely applied in many areas only requiring simple detections, like parking gate, speed measurement and so on. The authors applied the microwave sensors to landing platforms, which in my opinion is like moving the sensor from parking gate to a new location, without developing any new functions or creating any new concepts. The basic idea and application is the same as the sensor in any other locations. Even without considering these above, the hardware configuration is also very basic, excluding any advanced or customized functions. For example, if a bird flying through the detection area, could your device distinguish it from human? If the worst situation happened, like someone falling from heights, could your device detect it and call the ambulance immediately? Or at least record what happened at that time? The authors follow the widely applied ideas and functions of microwave sensors, simply applying it into landing platforms, without any interesting innovations or improvements. Personally I think it is lack of novelty.
- Even lack of novelty, if the author can demonstrate your solution is the best for a special situation, like landing platforms here, your paper is still attractive and valuable. However, the authors did not mention any other solutions. In my opinion, security cameras with computer vision algorithms processed by a single-board computer or cloud servers can do much better and more works. A wide vision camera may be enough to cover the two landing platforms and warn people who access the danger areas you plot in Figure 6. It can distinguish animals to people, can count how many people in the platforms, and record what they are doing. A stereo camera system even could map the positions of each person and warn him if close to any edge of the platforms. The cloud service can call the ambulance and project managers immediately if detecting anyone falling from the heights. What I mentioned above are all established techniques and have been applied in industry.
- Similar as second comment, I think the manuscript is lack of sufficient background and does not include all relevant references. After carefully reading the instruction section, I did not see any previous work or reports related to your proposal. What are the other solutions for FFH? Are you the first and the only one providing a solution for FFH? What is the problem not solved before? What are the advantages of your solution? Please clearly discuss them in your introduction section.
- The authors call their system “ambient intelligence”, which I cannot agree. As mentioned above, I did not see any intelligence in the system. Even for moving objects, you did not demonstrate it can distinguish birds from human, or count how many people in the area. And except sound alarm, the proposed system cannot do any further actions, which makes the application is very limited.
- Data analysis. I believe the sound warning will help workers to avoid the fall hazard area, however there are defects in your experiment design and data collection & analysis. For example, you did not demonstrate there was the same work load between two weeks in the platforms. Maybe people worked more in the danger area due to the project requirement in the first week, and they worked less in the next week because most work done. Also you did not trace the effectiveness of your proposal for a long time. People usually feel interesting to new things and pay more attention. While after weeks or months, they may be immunized from the sound and perform the same as before without the system. Additionally, the counted number of people may be inaccurate because your system cannot distinguish how many people in the area and only count as one. Your experiment data cannot strongly support your conclusion and answer my questions.
- I cannot agree some statements in your paper. Line 105 “However, it is impossible to continually monitor site operations in order to detect unsafe behaviors and/or conditions.” Personally I think it possible to monitor site operations by cameras + computer vision, detecting some unsafe conditions. Line 109 “However, the majority of these technologies rely on the fact that the worker will always need to wear the sensor.” I do not think the major techniques require workers to wear sensors to detect whether they are in fall hazard areas. The wearable technologies usually are used to detect biological factors of human.
Author Response
Comment 2.1 I very appreciate the authors’ effort to help constructors to reduce the risk of falling from heights (FFH). And I believe this system can warn people who access some danger areas, and reduce the potential risks. If it was an undergraduate student project demo in an exhibition, I definitely will recommend it. However, as a research paper, it looks not prepared well to be published. First of all, this paper is lack of originality and novelty. The microwave sensors (or Radar in generally) have been widely applied in motion detection. It was one of the dominant technologies to detect people access a certain area decades years ago before security cameras and computer vision technology developed. And still be widely applied in many areas only requiring simple detections, like parking gate, speed measurement and so on. The authors applied the microwave sensors to landing platforms, which in my opinion is like moving the sensor from parking gate to a new location, without developing any new functions or creating any new concepts. The basic idea and application is the same as the sensor in any other locations. Even without considering these above, the hardware configuration is also very basic, excluding any advanced or customized functions. For example, if a bird flying through the detection area, could your device distinguish it from human? If the worst situation happened, like someone falling from heights, could your device detect it and call the ambulance immediately? Or at least record what happened at that time? The authors follow the widely applied ideas and functions of microwave sensors, simply applying it into landing platforms, without any interesting innovations or improvements. Personally I think it is lack of novelty. Response 2.1 We are grateful to the reviewer for taking his/her time and effort to carefully read the manuscript, and especially for providing the detailed constructive comments and highlighting the major weaknesses in the research presented. For this first comment, we agree with the reviewer that the sensor technology used in this study is not new and innovative, and we are well aware of its general application in simple detections. We would like to explain that our intention for considering such rudiment detection technology was mainly because we needed a simple technology that can be easily developed and deployed onsite. In the context of a developing country (which is the aim of this special issue), sourcing of a robust and readily available technology, which is often considered rudimentary, is essential for the industry, especially small and medium enterprises. Initially, we considered a more advanced technology such as computer vision that allowed us to perform image analysis and enabled the incorporation other more sophisticated functionality. However, after some discussion with the practitioners, it was recommended that we consider a simpler technology that is more robust when installed at the construction site to achieve the aim of the study. Our aim of this study was primarily to examine how a simple AmI could be used to effect changes in safety behaviour of workers at a construction site. We hope that the simplicity of the technology will not deter the industry to invest their time and resource in improve site safety, especially In Thailand where the investment in safety equipment is relatively low.
2
We apologise that our aim does not offer a cutting-edge study that is in the forefront of innovations in construction safety. We admit that this a significant limitation of our study. Still, we do believe that our study demonstrates the potential of a simple technology in the prevention of onsite accidents, which could be of interest to those from developing countries where the access to advanced technologies is limited. In line with our response, we have added the above explanations in the manuscript to further clarify the aim and position of our study under Section 2.1. Comment 2.2 Even lack of novelty, if the author can demonstrate your solution is the best for a special situation, like landing platforms here, your paper is still attractive and valuable. However, the authors did not mention any other solutions. In my opinion, security cameras with computer vision algorithms processed by a single-board computer or cloud servers can do much better and more works. A wide vision camera may be enough to cover the two landing platforms and warn people who access the danger areas you plot in Figure 6. It can distinguish animals to people, can count how many people in the platforms, and record what they are doing. A stereo camera system even could map the positions of each person and warn him if close to any edge of the platforms. The cloud service can call the ambulance and project managers immediately if detecting anyone falling from the heights. What I mentioned above are all established techniques and have been applied in industry. Response 2.2 We agree with the suggestion from the reviewer that the CCTV technology typically used in security monitoring with image-based motion detection could also used to warn workers when they move into the designated hazardous areas. In fact, we did consider such technology in the early stage of the research. However, we found that motion detection using image detection in typical security cameras available in the market is quite sensitive to whatever object moving across the view of the camera. Given the cameras would have to be installed in the construction site environment, we were concern that dust, particles, and objects would interfere with the performance of the cameras. We therefore decided to use microwave sensors with simple audio alarm as we believe it would provide us with a more robust system. Despite knowing that we would have to give up many possible other functionalities, we believe that the selected approach should be sufficient to achieve our research aim. We have added the above justification to the manuscript to clarify our choice of technology employed in this study (under Section 2.1). We have also added the possibility of using stereo camera system to the future work section (under Section 5). Comment 2.3 Similar as second comment, I think the manuscript is lack of sufficient background and does not include all relevant references. After carefully reading the instruction section, I did not see any previous work or reports related to your proposal. What are the other solutions for FFH? Are you the first and the only one providing a solution for FFH? What is the problem not solved before? What are the advantages of your solution? Please clearly discuss them in your introduction section.
3
Response 2.3 During the literature review, we did not find past research that empirically examine how AmI could have an impact on construction workers’ safety behaviour. What we have come across, although not directly on the application AmI, are various publications on the application of sensors in construction safety to monitor workers and to develop smart construction site. Therefore, we were interested to investigate the behavioural aspect of the impact from the application of such technology. We have expanded our introduction part to include our justification for the research and the review of past research that involved the application of sensors in construction safety (See Section 1.3). Comment 2.4 The authors call their system “ambient intelligence”, which I cannot agree. As mentioned above, I did not see any intelligence in the system. Even for moving objects, you did not demonstrate it can distinguish birds from human, or count how many people in the area. And except sound alarm, the proposed system cannot do any further actions, which makes the application is very limited. Response 2.4 We admit that the intelligence part of our device is very limited to a sound alarm. It was also built for a very specific purpose to record the traffic of people moving past certain areas without the ability to distinguish between human or birds. As mentioned earlier, our aim was primarily to examine how a simple AmI could affect positive change in workers’ safety behaviour at a construction site. We also intended to keep the AmI system simple and durable that it can be easily developed and use at a construction site without much maintenance. We apologise that we could only address this in our manuscript under research limitation (see Section 5). Comment 2.5 Data analysis. I believe the sound warning will help workers to avoid the fall hazard area, however there are defects in your experiment design and data collection & analysis. For example, you did not demonstrate there was the same work load between two weeks in the platforms. Maybe people worked more in the danger area due to the project requirement in the first week, and they worked less in the next week because most work done. Also you did not trace the effectiveness of your proposal for a long time. People usually feel interesting to new things and pay more attention. While after weeks or months, they may be immunized from the sound and perform the same as before without the system. Additionally, the counted number of people may be inaccurate because your system cannot distinguish how many people in the area and only count as one. Your experiment data cannot strongly support your conclusion and answer my questions. Response 2.5 Regarding the uniform work load issue, we in fact were concern about this and did consult with the project management when chosing the appropriate period in which we can conduct the experiment. The appropriate period refers to the period that best represents the normal workload of the work for that particular floor of the building. Given the building with which
4
we conducted the experiment was level 14, the project manager had already had a good understanding of the normal workload period. We also acknowledge the lack of longitudinal experiment to fully investigate the long-term performance of this system. There were several limitations that prevented us from an extended experiment. We admit that this is the main weakness in our experiment that we have addressed in our research limitation (see Section 5). Comment 2.6 I cannot agree some statements in your paper. Line 105 “However, it is impossible to continually monitor site operations in order to detect unsafe behaviors and/or conditions.” Personally I think it possible to monitor site operations by cameras + computer vision, detecting some unsafe conditions. Line 109 “However, the majority of these technologies rely on the fact that the worker will always need to wear the sensor.” I do not think the major techniques require workers to wear sensors to detect whether they are in fall hazard areas. The wearable technologies usually are used to detect biological factors of human. Response 2.6 The former statement has been rewritten as “However, it is impossible to continually monitor site operations to detect unsafe behaviours and/or conditions without the help of technologies”. The latter statement has been removed and relevant paragraph modified.
Reviewer 2 Report
The submitted article is interesting but can be published after the following changes:
- Introduction contains so many old and irrelevant studies, it must be modified by adding new and relevant studies. Some of them are found by this reviewer and more can be found by authors. https://doi.org/10.1061/(ASCE)SC.1943-5576.0000465; https://doi.org/10.1061/(ASCE)SC.1943-5576.0000483; https://doi.org/10.1061/(ASCE)SC.1943-5576.0000468; https://doi.org/10.1061/(ASCE)SC.1943-5576.0000469.
- When we first reach abbreviation like ANOVA, it must be defined. This is not done at many places.
- The abstract is clear but not inline with methodology and other discussion. It must be the summary of whole study.
- The impact, objectives and expected outcome is not given clearly.
- Figures and Tables are not in good quality to understand.
- I use avoid giving references in conclusions, they could be discussed earlier and in Conclusions section only specific new conclusions must be listed.
Author Response
Comment 2.1 Introduction contains so many old and irrelevant studies, it must be modified by adding new and relevant studies. Some of them are found by this reviewer and more can be found by authors. https://doi.org/10.1061/(ASCE)SC.1943-5576.0000465; https://doi.org/10.1061/(ASCE)SC.1943-5576.0000483; https://doi.org/10.1061/(ASCE)SC.1943-5576.0000468; https://doi.org/10.1061/(ASCE)SC.1943-5576.0000469. Response 2.1 We would like to thank the reviewer for his/her feedback that has helped to improve the quality of our manuscript. We have expanded our introduction part to include more review of recent research that involved the application of sensors in construction safety (see Section 1.3). Comment 2.2 When we first reach abbreviation like ANOVA, it must be defined. This is not done at many places. Response 2.2 The full version of the ANOVA abbreviation are now provided in the abstract and in the paragraph where the abbreviation is used for the first time in the manuscript. Comment 2.3 The abstract is clear but not in line with methodology and other discussion. It must be the summary of whole study. Response 2.3 The abstract has been revised to better summarise the study. Comment 2.4 The impact, objectives and expected outcome is not given clearly. Response 2.4 The impact, objective and expected outcomes have been provided more clearly in the introduction section (under Section 1.3) Comment 2.5 Figures and Tables are not in good quality to understand. Response 2.5 We have improved the quality of the tables and figures. Comment 2.6 I use avoid giving references in conclusions, they could be discussed earlier and in Conclusions section only specific new conclusions must be listed.
2
Response 2.6 The statements with references have been moved to the discussion part. The conclusion has been modified to better provide the conclusion of the research. The limitations and recommendations for future research have also been added.
Reviewer 3 Report
Minor spelling and language corrections are recommended.
Author Response
Comment 3.1 Minor spelling and language corrections are recommended. Response 3.1 We would like to thank the reviewer for noting such issue. The paper has been proofread and corrections have been made to improve the readability of the paper.
Reviewer 4 Report
The paper must be improved before being published. The application is very important; However, it does not provide a contribution in the state of the art. As suggestion, it is necessary to add intelligense to the sensor node. As was demonstrated by the authors, currently the sensor node is gattering important data measurements that can be used for training artitial neural networks in order to propose an smart sensor.
Author Response
Comment 4.1 The paper must be improved before being published. The application is very important; however, it does not provide a contribution in the state of the art. As suggestion, it is necessary to add intelligence to the sensor node. As was demonstrated by the authors, currently the sensor node is gathering important data measurements that can be used for training artificial neural networks in order to propose a smart sensor. Response 4.1 We sincerely thank the reviewer for highlighting the improvement that can be further made to the study. We agree that enhancing the methodology to gather important data from the sensors that allow further analysis using such technique as ANN will be of immense benefits. For this particular study, the main purpose was to investigate how the basic form of AmI could effect changes in safety behaviour of onsite workers. Therefore, we kept our focus only on analysing the count data from the pre and post-AmI implementation to examine such changes. In addition, it was our intention that the AmI must be simple and robust so it can be used at a construction site. Therefore, the sensors used in the present study had no advanced capability to collect other types of data (e.g. image and temperature detection). We do acknowledge such limitation and have now included in the paper to inform the readers regarding this (see Section 5).
Round 2
Reviewer 1 Report
I appreciate the authors’ further effort to polish your manuscript according the previous reviews, although you did not address any major issues I commented last time.
Honestly speaking, this paper does not achieve my standard for publication. In technology part, your device is very simple and the function is quite basic, without any advanced or customized functions. Besides I cannot agree with your reply that “motion detection using image detection in typical security cameras available in the market is quite sensitive”. It is not a research that simply purchasing a camera from market without any customized functions. Actually, lots of security cameras include face/human detection instead of the simplest motion detection, which could filter the unnecessary detections of particles, birds, cars and others. In field application part, your experiment design is too simple to address my concerns in the last review. And it looks you cannot improve the experiment anymore.
However, I could accept this manuscript conditionally if it was designed for a special issue regarding developing countries. I understand that the investment in safety equipment in developing countries is relatively low, and small companies usually focus on cutting cost, avoiding any expensive safety devices. I very appreciate to see the authors’ effort to help workers in developing countries although the devices and the technology may be not advanced. It’s better than nothing.
Reviewer 2 Report
Although the revised is improved but still I think the introduction and conclusions section can be improved further. Some of the article were suggested not only to add them but to show that there is need to check for recent advancement. I have had again search and find some relevant studies: Qian, J., Li, J., Li, C., Zhou, J., & Yao, G. (2019). Stress Measurement of Steel Strands Based on System Identification Method. Shock and Vibration, 2019.; System Identification of Code Conforming Low-Rise RC Building in Kathmandu, Nepal; R Dhakal, R Rupakhety, G Dipendra, S. E. Rahimi; XI International Conference on Structural Dynamics; Yang, Y., Chen, J., Lan, F., Xiong, F., & Zeng, Z. (2018). Joints parameters identification in numerical modeling of structural dynamics. Shock and Vibration, 2018.; Hana’a M. Alqam, Anoop K. Dhingra, "Motion Transmissibility for Load Identification Based on Optimum Sensor Placement", Shock and Vibration, vol. 2019, Article D 7810686, 13 pages, 2019. https://doi.org/10.1155/2019/7810686.; Gang Yu, "An Underdetermined Blind Source Separation Method with Application to Modal Identification", Shock and Vibration, vol. 2019, Article ID 1637163, 15 pages, 2019. https://doi.org/10.1155/2019/1637163; Peng Wen, Inamullah Khan, He Jie, Chen Qiaofeng, Yang Shiyu, "Online Intelligent Identification of Modal Parameters for Large Cable-Stayed Bridges", Shock and Vibration, vol. 2020, Article ID 2040216, 17 pages, 2020. https://doi.org/10.1155/2020/2040216
Reviewer 4 Report
The paper was improved; however, it is necessary to increase the contribution in the state of the art, in the future. However, this reviewer recognizes the effort of the authors.